# Chronic Heat Stress Can Induce Conjugation of a Novel *ermB*-Containing ICE_FZMF_, Increasing Resistance to Erythromycin Among *Enterococcus* Strains in Diverse Intestinal Segments in the Mouse Model

**DOI:** 10.3390/antibiotics14050460

**Published:** 2025-04-30

**Authors:** Lingxian Yi, Zining Ren, Yu Feng, Yechun Zhang, Jianshuo Liu, Xiaowu Yuan, Qihong Kuang, Hui Deng, Bo Yang, Daojin Yu

**Affiliations:** Fujian Key Laboratory of Traditional Chinese Veterinary Medicine and Animal Health, College of Animal Sciences, Fujian Agriculture and Forestry University, Fuzhou 350002, China; lingxian_yi@fafu.edu.cn (L.Y.); 18233167971@163.com (Z.R.); fyu62522@outlook.com (Y.F.); zhangyechun0602@outlook.com (Y.Z.); l15216592650@163.com (J.L.); 13606908177@163.com (X.Y.); kuangqh0816@126.com (Q.K.); deng12345hui@163.com (H.D.); ybvet1@fafu.edu.cn (B.Y.)

**Keywords:** heat stress, antimicrobial resistance, *Enterococcus* isolates, mobile genetic elements, one health

## Abstract

Background: The impact of heat stress on intestinal bacterial antimicrobial resistance (AMR) and its underlying mechanisms is not fully understood. This study aims to explore how heat stress influences AMR in the gut and the mechanisms involved. Methods: A Specific-Pathogen-Free (SPF) mouse model was used, divided into a control group (maintained at 25 °C) and a heat stress group (exposed to 42 °C for 30 min twice daily for 55 days). The effectiveness of the model was verified by RT-qPCR and histopathological analysis. Antibiotic susceptibility testing and clonal analysis (ERIC-PCR) were performed. Colonization assays were conducted to determine the accumulation of resistant strains in the gut. Metagenomic sequencing was conducted to investigated microbial composition. Results: RT-qPCR and Histopathological analysis revealed intestinal damage and significant upregulation of genes related to stress response, intestinal barrier integrity and inflammation, indicating successful model establishment and physiological alterations. Antibiotic susceptibility testing revealed increased resistance to erythromycin, chloramphenicol, and tetracycline among *Enterococcus* strains. Clonal analysis demonstrated that these resistant strains were clonally unrelated. Sequencing identified a novel *ermB*-carrying integrative and conjugative element (ICE_FZMF_) among four erythromycin-resistant strains. The rectum harbored a higher proportion of erythromycin-resistant *Enterococcus* strains with elevated minimum inhibitory concentrations (MICs) after 25 days of heat stress exposure. Colonization assays confirmed that heat stress led to the accumulation of erythromycin-resistant *Enterococcus* in the rectum. Metagenomic sequencing revealed significant changes in microbial composition, favoring anaerobic metabolism. Conclusions: This study suggests that chronic heat stress can promote the emergence of antibiotic-resistant strains through ICE transfer, providing insight for environmental safety.

## 1. Introduction

With rising global temperatures and increased breeding density, livestock are frequently challenged by heat stress. In high-temperature environments, livestock regulate their body temperatures by reducing feed intake and increasing metabolic activity [1]. This subsequently leads to reduced feed efficiency, decreased weight gain, compromised immunity and disruptions in the microbiota ecosystem in the intestine and its metabolism [1]. However, the impact of heat stress on the emergence, persistence and transmission of antimicrobial-resistant bacteria in the gut remains underexplored.

In the 21st century, bacterial antimicrobial resistance has emerged as a significant challenge to human health [2]. The World Health Organization (WHO) has endorsed the “One Health” approach to address this complex issue, emphasizing the need to consider environmental, clinical and livestock-related factors [3]. The emergence and spread of antimicrobial-resistant bacteria are driven by various factors, including environmental pollution and the use of antimicrobials in clinical and livestock settings [4]. Here, we hypothesize that environmental heat stress may influence the development of antibiotic resistance in intestinal microbiota.

Enterococci are widely prevalent opportunistic pathogens, playing a crucial role in transmitting pathogenic and multidrug-resistant strains in livestock farming and food processing [5,6]. They are frequently used as indicators for antibiotic resistance among Gram-positive bacteria in the intestine [5]. The resistance and spread of enterococci in clinical settings have become major concerns. Previous research has revealed severe resistance to erythromycin, vancomycin, chloramphenicol and tetracycline among enterococci [7]. Since 2015, resistance rates to erythromycin and tetracycline in *Enterococcus faecium* isolated from livestock farms in China have been over 90% [8]. By 2022, erythromycin-, tetracycline- and vancomycin-resistant enterococci accounted for approximately 60% of all clinical samples, making them the most prevalent multidrug-resistant strains [7]. Among these, the *ermB* gene is the primary determinant of erythromycin resistance. Our previous research demonstrated that heat stress enhances the occurrence of erythromycin-resistant *Enterococcus* isolates in mouse feces, predominantly mediated by the *ermB* gene [9]. However, the mechanisms underlying heat stress-induced erythromycin resistance remain unclear.

Previous studies primarily focused on the direct effects of heat stress on bacterial strains in vitro. Heat stress would directly reduce antibiotic binding rates in *Salmonella* Typhimurium, thereby increasing resistance to cefotaxime, tetracycline, and ampicillin [10]. Similarly, heat stress has been shown to be associated with antibiotic resistance in emerging foodborne pathogens, such as *Cronobacter sakazakii* and induced antibiotic resistance to macrolides in *Lactococcus lactis* in vitro [11,12]. In addition, heat stress reduced the sensitivity of *Escherichia coli* to rifampin in vitro [13]. However, there is a gap in the research on the effects of environmental heat stress on the antimicrobial resistance of bacteria in animal intestines.

In this study, we used Specific-Pathogen-Free (SPF) mice to establish a heat stress model to observe the damage and changes in the intestinal microbial environment, antibiotic resistance of *Enterococcus* strains from different intestinal segments and molecular mechanisms involved in the development of antibiotic resistance in *Enterococcus* strains due to heat stress. This research aimed to elucidate the critical role of environmental temperature changes in the development of antimicrobial-resistant bacteria, providing reference for reducing bacterial resistance and improving animal living and transportation conditions.

## 2. Results and Discussion

### 2.1. Validation of Heat Stress Model via Molecular Markers of Intestinal Barrier Disruption and Inflammation

Markers of intestinal stress, inflammation and function were assessed at days 7, 15, 30 and 45. Heat shock proteins (HSPs) are key markers of the heat shock response in biological systems [14]. HSP27 is involved in cell death pathways and intestinal epithelial responses [15]. In the HS group, HSP27 expression peaked on day 7, with a 14.7-fold increase in the ileum, 12.2-fold increase in the cecum and 3.7-fold increase in the rectum, while the colon reached its highest level on day 15 with a 12.2-fold increase (Appendix A). Elevated levels persisted through later stages, showing a 2.1-fold increase in the ileum and 1.7-fold increase in the colon by day 45, suggesting its protective role in maintaining intestinal barrier integrity under heat stress.

HSP70, a crucial protein in the HSP family, has been shown to enhance cell survival under stress by increasing cellular tolerance to adverse internal and external conditions [15]. Compared with the CON group, HSP70 expression was highest during the early stress phase and gradually declined over time in the cecum and rectum of the HS group (Appendix A). HSP70 mRNA expression also peaked on day 7 after heat stress, with 200-fold increased expression in the ileum, a 9.5-fold increase in the cecum, a 32-fold increase in the colon and a 43-fold increase in the rectal. Although HSP70 expression in the HS group decreased after day 7, it remained higher than that in the CON group. The decreased trend after day 7 indicated a potential decline in its regulatory capacity during later stages of heat stress.

HSP90 plays an important role in facilitating epithelial cell repair in the small intestine [16]. The trends in HSP90 mRNA expression varied slightly among the segments in the HS group (Appendix A). Expression increased by 5.4-fold on day 7 and was significantly upregulated by 3.4-fold on day 15. The highest expression levels in the cecum, colon and rectum were observed on day 30, with increases of 8.2-fold, 1.6-fold and 4.3-fold, respectively. These findings are consistent with previous reports suggesting that HSPs play a crucial role in cellular protection and stress response by facilitating protein folding and protecting cells from damage during heat exposure [17]. Additionally, these results confirm the successful establishment of the heat stress model in mice.

A similar trend was observed in the mRNA abundance of stress and integrity gene markers. The mRNA abundance for the tight junction protein claudin 3 (*cldn3*) exhibited significant increases during the early stress stages. In the ileum, expression rose 14-fold on day 7 and 4.9-fold on day 15. In the cecum, expression increased 4.3-fold, while in the colon, it rose 3.5-fold by day 15. Rectal Cldn3 expression peaked at a 2.8-fold increase on day 30, followed by a decline in the later stress stages (Appendix A). An increasing trend in the Cldn2 mRNA levels was observed in all four intestinal segments during the early stages of stress (Appendix A). By the later stages of heat stress, significant differences in Cldn2 expression were observed in the ileum and rectum compared with the CON group, while no significant differences were found in the cecum and colon. This result is consistent with previous studies indicating that the body can modulate the expression of intestinal tight junction proteins, thereby influencing intestinal permeability under heat stress [18]. However, the expression of tight junction proteins varies across different intestinal segments [18,19].

Cryptdins, a class of antimicrobial peptides, have been shown to inhibit the growth of *Escherichia coli* and *Salmonella* spp. [20]. The elevated expression of Cryptdins plays a crucial role in maintaining intestinal barrier integrity, while reduced levels can impair the function of the intestinal mucosal immune system, decreasing resistance and increasing the risk of bacterial invasion, which can trigger inflammatory responses in the host [20]. In this study, the *crypt-1* gene expression in the HS group initially decreased and subsequently recovered. In the ileum, cecum and rectum, *crypt-1* gene expression remained high until day 15 but dropped significantly after day 30, with partial recovery by day 45. In the colon, *crypt-1* gene mRNA levels increased 33-fold on day 7 and 1.8-fold by day 45 (Appendix A).

Heat stress has been shown to disrupt intestinal integrity through inflammatory cytokines, such as interferon-γ, TNF-α, IL-1β, IL-4, IL-6 and IL-13, which activate MLCK and increase intestinal permeability [21]. In this study, IL-6 mRNA levels varied across intestinal segments during heat stress. In the ileum, expression increased by 2.4–10-fold over 45 days, except on day 30. In the cecum, expression peaked at 6-fold on day 30, while colonic levels rose steadily until day 30 before declining to 2.8-fold by day 45. Rectal IL-6 expression peaked during the first 15 days, with 4.3-fold and 3.9-fold increases on days 7 and 15, respectively (Appendix A). TNF-α mRNA expression was elevated during the early stages of stress but decreased in the later phases. In the ileum, expression increased 10-fold on day 7 and 9-fold on day 15, followed by a 3–4-fold rise in the later stages of heat stress. In the cecum, expression peaked at 8.5–8.9-fold during the early stages and then decreased. Colonic expression surged 32–38-fold in the early phases, followed by a 2–3-fold increase in later stages. Rectal TNF-α expression increased 10-fold on day 7, 21-fold on day 15, and 4-fold on day 30, with all changes showing significant differences from the controls (Appendix A).

Overall, the different trends observed in the mRNA abundance of markers for intestinal stress and integrity suggest that the intestinal segments exhibited diverse tolerance to heat stress, with some segments showing persistent inflammation.

### 2.2. Histopathological Validation of Heat Stress-Induced Intestinal Damage in Mice

The intestine not only serves as a key organ for nutrient absorption but also plays a vital role as a protective barrier against pathogens [22]. When the intestinal barrier is compromised, this can lead to diseases affecting other organs, including metabolic disorders, liver disease and obesity [23,24]. Previous studies have shown that heat stress causes significant damage to the intestinal tract [25].

To further validate the physiological relevance of our heat stress model, we observed structural damage to the intestinal villi, necrosis of epithelial cells, inflammatory cell infiltration and signs of congestion and hemorrhage in the ileum, cecum, colon and rectum on days 15 and 30 of heat stress (Figure 1). Additionally, lymphoid follicle hyperplasia and glandular atrophy were noted, further confirming that heat stress caused structural damage to these intestinal segments. It has been reported that this damage would impair the normal digestive and absorptive functions of the intestine, increase intestinal permeability and promote bacterial translocation, potentially leading to systemic infections [26]. The damage to the intestinal lining and the occurrence of bacterial translocation may influence the development of antibiotic resistance in gut bacteria, which requires further confirmation.

### 2.3. Determination of MICs for Enterococcus Strains

During the 55-day heat stress exposure, no significant changes in the MIC values were observed for ampicillin, vancomycin, rifampin or ciprofloxacin in any of the intestinal sections (Appendix A). However, heat stress induced significant increases in the MIC values for chloramphenicol, tetracycline and erythromycin, with resistance rising over time across all intestinal regions (Figure 2A).

The change in erythromycin resistance is obvious. The prevalence of erythromycin-resistant *Enterococcus* strains in all intestinal segments was detected by using selective media containing erythromycin (4 μg/mL). No resistant strains were detected in the CON group throughout the 55-day study. In the HS group, resistant strains were all present after day 15, with detection rates of 0.002%, 0.03%, 0.064% and 0.26% in the ileum, cecum, colon and rectum, respectively, and the resistant strains were more frequently isolated from the hindgut (colon and rectum) than the foregut (ileum and cecum), especially in the rectum (Figure 2B). The rectum consistently exhibited the highest detection rates compared with other intestinal segments, with an erythromycin resistance rates of 0.26%, 0.24%, 0.066% and 0.0323% at 15, 30, 45 and 55 days, respectively. The rates are listed in Appendix A in detail. The results are different from the development of antibiotic resistance observed within *E. coli* in pigs. It has been reported that heat stress increases the resistance of *E. coli* to ampicillin and tetracycline in the hindgut compared with the foregut, which is associated with heat stress accelerating the movement of *E. coli* in the intestines [27].

A total of 4–10 colonies from plates without erythromycin were randomly selected. In total, 154 strains from the CON group and 225 isolates from the HS group were used to determine the MICs of seven antibiotics. We found that all *Enterococcus* strains in the CON group were all sensitive to erythromycin (MIC < 4 μg/mL), while the HS group showed an increased MIC value for erythromycin after day 15 in the ileum, cecum, colon and rectum (Figure 2(Cb)). Notably, on days 25, 45 and 55 of heat stress, the HS group exhibited an increasing trend in the erythromycin MIC along the intestinal tract. At day 15, resistant enterococci (MIC ≥ 8 μg/mL) initially appeared in the ileum and cecum of the HS group (Figure 2(Cb)). By day 25, high-level resistance (MIC ≥ 128 μg/mL) was detected in enterococci from the ileum, cecum, colon and rectum, with the erythromycin MIC of isolates from the rectum being significantly higher than those from other intestinal segments (Figure 2(Cc–Cf)), indicating that the MIC of erythromycin increased along the intestinal segments.

Excluding erythromycin, although tetracycline- and chloramphenicol-resistant isolates were also detected in the CON group, their overall MIC distributions were lower than those observed in the HS group. On days 25 and 45 of heat stress, the MIC of chloramphenicol for enterococci isolated from the ileum, colon and rectum was significantly higher than that in in the CON group. Similarly, on days 25 and 35, enterococci isolated from the cecum showed a markedly increased chloramphenicol MIC compared with the controls. Furthermore, on day 25, the tetracycline MIC for enterococci from all intestinal segments was significantly elevated relative to the CON group, and on day 35, isolates from the ileum and rectum continued to exhibit a significantly higher tetracycline MIC than those from the CON group (Figure 2(Ab,Ac)).

### 2.4. Antibiotic Resistance Gene (ARG) Screen and Clonal Relatedness Analysis

Based on the MIC results, we then screened 154 strains from the CON group and 225 isolates from the HS group for erythromycin resistance genes, including *ermA*, *ermB* and *mefA*, chloramphenicol resistance genes *fexA* and *optrA* and tetracycline resistant genes *tetS* and *tetL*. Detailed results are provided in Table 1. No *ermA*, *ermB*, *mefA*, *fexA* or *optrA* genes were detected in any of the strains from the CON group.

In the HS group, the erythromycin resistance gene *ermB* was most prevalent, with a detection rate of 60.1%. The highest detection rate was found in isolates from the rectum at 76.7%, followed by those from the colon at 68.8%, the cecum at 58.2% and the ileum at 31.2%. The genes *ermA* and *mefA* were detected in fewer strains, primarily from the rectum and ileum, with detection rates of 0.9% and 0.5%, respectively. The results indicate that the erythromycin resistance in enterococci induced by heat stress is primarily mediated by the *ermB* gene, which is consistent with our previous observations in mouse feces [9].

The chloramphenicol resistance gene *fexA* was more prevalent than *optrA*, with detection rates of 4.9% and 1.3%, respectively. The highest detection rate for *fexA* was observed in isolates from the cecum at 10.9%, followed by the rectum at 5%, the ileum at 2.1% and the colon at 1.6%. The tetracycline resistance genes *tetS* and *tetL* were detected both in the CON and HS group, with *tetS* being more prevalent than *tetL*. No significant difference was observed between the CON group and the HS group. These results indicated an intestinal segment-specific distribution of antimicrobial resistance genes in response to heat stress.

To investigate the reasons for the emergence of erythromycin-resistant enterococci, 95 strains from the CON group and 121 isolates from the HS group were randomly selected for clonal typing by using ERIC-PCR. The results showed that all erythromycin-resistant strains were genetically unrelated, with genetic similarities below 90% (Appendix A). This indicates that the erythromycin-resistant strains were non-clonal and likely arose independently.

### 2.5. Characterization of a Novel ICE_FZMF_ and Its Transferability

Horizontal gene transfer (HGT) plays an important role in the development of antimicrobial resistance (AMR) [28]. The ARGs transfer globally through diverse mobile genetic elements, such as plasmids, integrative and conjugative elements (ICE), transposons, bacteriophages and genomic islands (GEIs) [29]. When we analyzed the chromosome sequence of FZMF (NO. CP127160), the whole genetic sequence of ER01 (PRJNA1231587), B1 (JAUKPL000000000), B2 (JAUKPJ000000000) and B3 (JAUKPK000000000), we found a novel 140 293 bp ICE element in the chromosome sequence of FZMF with an average GC content of 38.79%, designated ICE_FZMF_ (Figure 3A). The element contained nine antibiotic-resistance genes—*mef(A)*, *msr(D)*, *ant(6)-Ia*, *tet(L)*, *erm(C)*, *erm(B)*, two copies of *tet(S)* and *aac(6′)-E526*—inserted into an RNase J encoding gene, with a direct repeat of 5′-aaaaacggcgatttt-3′ in the flanking region (Figure 3A). An 18 334 bp size composite transposon organized in the IS*Lgar5*-*tet(L)*-*ermC*-*ermB*-*tet(S)*-IS*Lgar5* structure was located on the ICE_FZMF_. Inverse PCR and sequencing with IS-F/R primers targeting the junction revealed the formation of a circular intermediate of the *ermB*-containing translocatable unit (Figure 3C). Additionally, a perfect 50-bp inverted repeat sequence IRL:5′-GAGAGTGTAAAATATTTTGTGTAAATAGAAAAAAGGAAGTCCCTTCTGTA-3′ and IRR:5′-GGGAGCGTCAATAATTTTGTGTAAATGATTCTTCCCTACTGCAGATTGTT-3′ was detected at the flanking region, which is a common feature among members of IS*Lgar5* (Figure 3A). In addition, the ICE contained *traF* and *virB4* genes, which encode components of the type IV secretion system (T4SS), indicating its transferability. Inverse PCR with CE-F/R primers targeting the junctions was performed, followed by Sanger sequencing, which verified the formation of a circular intermediate of ICE_FZMF_ (Figure 3C)_._ The sequence of the junction was uploaded to NCBI (Accession No. PRJNA1231627).

The transferability of ICE_FZMF_ was confirmed by a filter mating assay. The *ermB*-carrying ICE_FZMF_ was transferred from donor strains *Enterococcus casseliflavus* FZMF and *Enterococcus avium* ER01 to a recipient *Enterococcus faecalis* strain ES03 at frequencies of (3.73 ± 2.78) × 10^−9^ and (1.1 ± 0.7) × 10^−9^, respectively. The *ermB*-carrying ICE_FZMF_ can horizontally self-transfer by conjugation at a low transfer frequency, which supports the dissemination of the *ermB* gene across species boundaries.

We compared the sequence of ICE_FZMF_ with the separate contigs of B1 (JAUKPL000000000), B2 (JAUKPJ000000000), B3 (JAUKPK000000000) and ER01 with BLAST and then mapped them to ICE_FZMF_ with the Proksee planform. The four strains all contain a similar backbone of ICE_FZMF_, indicating that heat stress could induce the conjugation and integration of *ermB*-containing ICE_FZMF_ and cause the presence of the erythromycin-resistant enterococci (Figure 3B). However, several antibiotic resistance genes were missing. The ER01 strain carried only the *ermB* and *tet(M)* genes, and the B1 strain carried the *ermB*, *tet(M*), *tet(S)*, *optrA* and *ant(6)-Ia* genes. The B2 strain carried the *ermB*, *tet(L)*, *ant(6)-Ia* and *fexA* genes, and the B3 strain harbored the *ermB*, *tet(L)* and *ant(6)-Ia* genes. The selective pressure of erythromycin resulted in the consistent presence of the *ermB* gene, and the confirmation of the circular intermediate formation of IS*Lgar5*-*tet(L)*-*ermC*-*ermB*-*tet(S)*-IS*Lgar5* indicated the probable self-transferability of the *ermB* gene within strains [30].

### 2.6. Heat Stress Facilitated Enhanced Rectal Colonization of Erythromycin-Resistant Enterococcus ER01

To confirm the high prevalence of erythromycin-resistant enterococci isolated from the hindgut compared with the foregut, a single erythromycin-resistant *Enterococcus* strain, designated ER01, which carries the erythromycin resistance gene *ermB* and the tetracycline resistance gene *tet(M)*, was randomly selected as a marker strain. This strain was administered to mice via oral gavage for three consecutive days. After a one-day break, fecal samples from mice in both the CON and HS groups were randomly collected and plated onto SBM medium containing 256 μg/mL of erythromycin. The colonies that grew confirmed the successful colonization of ER01.

The time of 0 h was defined as the point before heat stress. At 0, 3, 6, 9 and 12 h, the intestinal tissues of five randomly selected mice were obtained. After appropriate dilution, liquid was plated on SBM medium containing 256 μg/mL of erythromycin. After incubation at 37 °C for 48 h, colonies present on the plates were counted and confirmed to be clonally related via ERIC-PCR fingerprint analysis. All erythromycin-resistant enterococci were genetically similar to ER01 with a genetic similarity of 100% (Appendix A). The results showed that at 6, 9 and 12 h after heat stress, the proportion of erythromycin-resistant enterococci colonizing the rectum was significantly increased, suggesting that the gut environmental change in the rectum was induced by heat stress, making it more favorable for the colonization and proliferation of erythromycin-resistant strains (Figure 2D). However, the factor induced by heat stress that leads to the preference of ER01 colonization in the rectum remains unclear. The intestine is frequently regarded as a reservoir for bacterial cultivation and the development of antibiotic resistance [31]. It is widely known that the intestinal microenvironment, composed of intestinal bacteria and its metabolic products, would change with the environment temperature in pigs and laying hens [32,33,34]. Hence, we then studied the effect of heat stress on the intestinal microenvironment in the mouse intestine.

### 2.7. Effects of Heat Stress on Microbiota Composition in Different Intestinal Segments of Mice

To assess the change in the microenvironment in the intestine, metagenomic sequencing and analysis were performed. The indices of alpha diversity showed that the HA (ileum before heat stress) and HB (ileum after heat stress) groups had the lowest and highest microbial diversity, respectively, across all three indices (Shannon, Simpson and InvSimpson), and the ZB group (rectum after heat stress) also showed slightly greater diversity than the ZA group (rectum before heat stress), indicating that the heat stress significantly increased the microbial diversity in both the ileum and rectum (Figure 4A).

As shown in the principal component analysis (PCA), principal coordinate analysis (PCoA) and multidimensional scaling (MDS) analysis, the results revealed significant differences in species composition and community structure between the HA and HB as well as ZA and ZB groups, especially ZA and ZB (Figure 4B), suggesting that heat stress alters species compositions. Anosim analysis was employed to test the significance of differences between and within groups. Based on the species-level Anosim results (Figure 4C), significant differences in community composition were shown between both HA and HB (R = 0.428, *p* = 0.026) as well as ZA and ZB (R = 0.588, *p* = 0.009), particularly for the latter.

The dominant bacterial phyla in both the ileum and rectum microbiota included Bacteroidetes (formerly known as Bacteroides), Firmicutes (previously referred to as Firmicutes) and Pseudomonadota (formerly known as Proteobacteria). Before heat stress, the ileum exhibited a higher proportion of Firmicutes and a lower proportion of Bacteroidetes compared with the rectum. After heat stress, the relative abundance of Firmicutes increased in the rectum and decreased in the ileum, while the relative abundance of Bacteroidetes decreased in the rectum and increased in the ileum. The differences in the proportions of Firmicutes and Bacteroidetes among groups suggest that heat stress significantly reshaped the microbial composition in different intestinal segments. It has been known that a lowered Firmicutes-to-Bacteroidetes (F/B) ratio in the ileum suggests inflammation, which is consistent with elevated expression of Crypt-1, IL-6 and TNF-α at the end of the heat stress exposure [35].

### 2.8. Metabolic Pathway Alterations in Response to Heat Stress

The results from the LEfSe analysis, performed with a linear discriminant analysis (LDA) score ≥ 2, revealed significant changes in the microbial metabolic pathways in response to the heat stress in both the ileum and rectum (Figure 5A).

In the ileum, heat stress led to an increase in pathways associated with stress adaptation and repair mechanisms, such as MicroRNA in cancer (ko05206) and pentose and glucuronate interconversions (ko00040). Notably, the HB group showed a marked upregulation of galactose metabolism (ko00052), indicating a shift in the microbial community toward energy production via sugar metabolism under heat stress. Conversely, pathways linked to organic carbon oxidation, such as protein processing in the endoplasmic reticulum (ko04141), were more abundant in HA, suggesting basal metabolic activity prior to heat exposure. Similarly, in the rectum, ZB (rectum after heat stress) exhibited an increased abundance of pathways related to carbon fixation and fermentation, such as galactose metabolism (ko00052) and acarbose and validamycin biosynthesis (ko00525), reflecting a shift toward energy production via fermentation and the assimilation of carbon under stress. In contrast, ZA (rectum before heat stress) primarily showed abundant pathways linked to protein processing and amino acid biosynthesis, highlighting a focus on growth and maintenance without heat stress.

The increased abundance of enzymes involved in galactose metabolism and related pathways suggests a transition to more anaerobic metabolic processes, enhancing the microbial community’s ability to cope with heat-induced stress [36]. This study underscores the dynamic nature of the gut microbiome’s metabolic flexibility and its potential role in promoting stress resilience. However, the direct factor for the conjugation and integration of ICE_FZMF_ requires further investigation.

### 2.9. Effects of Heat Stress on Antimicrobial Resistance Genes in Gut Microbiota

As shown in the heat map, the ARG abundance in the rectum was higher than that in the ileum, both with and without heat stress (Figure 5C). Overall, most ARG abundances decreased markedly after heat stress.

To identify resistance genes that exhibited significant differences between groups, we used Metastats to analyze the resistance gene abundance at various levels. This analysis yielded *p* values, which were then corrected to obtain *q* values. Genes with significant differences were visualized in bar charts (Figure 5D). In the ileum, a significant shift was observed for the *tetQ* gene in the HB group, while the abundance of tetracycline resistance genes *tetO* and *tet32*, as well as the bacitracin resistance gene *bacA*, increased in the HA group. In the rectum, ARGs such as the tetracycline resistance genes *tetW* and *tet40* and the vancomycin resistance gene *vanR* showed greater abundance in the ZB group, whereas the class A β lactamase resistance gene *bla* and the aminoglycoside resistance gene *aadE* were more abundant in the ZA group. Consistent with our previous reports, the abundance of the *ermB* gene was reduced within the bacteria in the mouse feces, which further support the results that the occurrence of erythromycin-resistant *Enterococcus* strains is mediated by the conjugation of *ermB*-containing ICE_FZMF_ from other bacterial species.

Overall, although the findings of our study were observed in a mouse model, similar effects may occur in farm animals. Chronic heat stress disrupts intestinal integrity and promotes the horizontal dissemination of antimicrobial resistance genes. These results highlight the importance of implementing effective heat stress mitigation strategies in animal husbandry to protect both animal welfare and public health. Various strategies have been employed in livestock production to alleviate the impact of heat stress. These include modifications to housing systems, nutritional interventions, breeding for heat-tolerant breeds and the practice of seasonal altitude migration [37,38]. These farm management strategies can help lower the environmental temperatures of animals, therefore reducing the risk of resistance gene transfer and promoting animal health.

## 3. Methods

### 3.1. Establishment of a Mouse Heat Stress Model

The mouse heat stress model was established by a previous method [9,39]. Briefly, a total of 100 SPF-grade female ICR (SLAC:ICR) mice weighing 18–22 g (3–5 weeks old) were randomly divided into a control group (CON group, *n* = 50) and a heat stress group (HS group, *n* = 50). Before the experiment, the mice were housed at a controlled temperature of 22–25 °C and 50% humidity for 7 days. During the experiment, food and water given to mice were sterilized. The CON group mice were maintained at 22–25 °C and 50% humidity. The HS group was subjected to heat stress by placing the mice in a temperature-controlled incubator set to 42 °C at 50% humidity (based on our previous experimental results) for 30 min in the morning and 30 min in the afternoon and then returned to 22–25 °C. The exposure was performed daily for 55 consecutive days.

### 3.2. Histological Analysis

Intestinal tissues from the ileum, cecum, colon and rectum were collected on days 0, 15 and 30 from 2 mice per group. The tissues were fixed in 4% paraformaldehyde for 18–24 h, rinsed at least three times with physiological saline and preserved in 70% ethanol. Fixed intestinal tissues were dehydrated in a series of graded ethanol baths (75%, 85%, 90%, 95% and 100%) followed by xylene three times to remove ethanol. The dehydrated tissue block was then embedded in paraffin wax. The tissue blocks were sectioned at a thickness of 5 μm and then mounted onto glass slides and baked at 55 °C for 10 min. For hematoxylin and eosin (H&E) staining, paraffin sections were deparaffinized twice in xylene, rehydrated in a series of graded ethanol baths (100%, 95%, 85% and 75%) for 5 min each and rinsed in distilled water. Slides were stained with hematoxylin for 3 min followed by distilled water rinse and then re-stained with eosin for 5 min. The stained slices were again subjected to dehydration with graded ethanol baths and xylene and cover slipped with mounting medium. Each section was analyzed under a light microscope at 100x magnification.

### 3.3. Detection of mRNA Expression Levels of Heat Shock Proteins, Markers of Intestinal Stress, Integrity and Inflammatory

Five mice per group were aseptically sacrificed at days 7, 15, 30 and 45, and tissues were collected from the ileum, cecum, colon and rectum for qPCR analysis. The total RNA was extracted from the tissues while following the instructions of the MagicPure^®^ Total RNA Kit (TransGen Biotech, Beijing, China). The extracted RNA was reverse transcribed into cDNA while following the instructions of the TransScript^®^ First-Strand cDNA Synthesis SuperMix Kit (TransGen Biotech). Using β actin as the reference gene, qPCR was performed to detect the mRNA expression levels of heat shock proteins (HSP27, HSP70 and HSP90), tight junction proteins (Cldn2, Cldn3), cryptdin (Crypt-1), interleukin-6 (IL-6) and tumor necrosis factor-α (TNF-α) in the intestinal tissues, with the primers listed in Appendix A. Histological sections were also prepared to observe damage to different intestinal segments.

### 3.4. Sampling and Isolation of Intestinal Tissues and Bacteria

The intestinal contents from the ileum, cecum, colon and rectum were aseptically collected from 3 mice per group on days 0, 15, 25, 35 and 45 for bacterial isolation and minimum inhibitory concentration (MIC) testing. The collected samples were mixed with 1 mL of 0.85% saline. The mixture was vortexed and serially diluted with 0.85% saline, and 20 μL of the appropriately diluted suspension was plated on *Enterococcus* selective medium, in this case Slanetz & Bartley Medium (SBM, Beijing Solarbio Science & Technology Co., Ltd., Beijing, China), containing erythromycin at concentrations of 0 μg/mL or 4 μg/mL. After 24 h of incubation at 37 °C, the colonies present on the plate with and without 4 μg/mL erythromycin were counted to calculate the erythromycin resistance rates. From 4 to 10 single colonies were randomly picked from plates with 0 μg/mL and 4 μg/mL erythromycin for further *Enterococcus* strain identification and MIC determination. Genomic DNA of the isolated strains was extracted using a bacterial DNA kit (D3350-01, Omega Bio-tek, Norcross, GA, USA), and the 16S rRNA primers listed in Appendix A were used to identify *Enterococcus* strains.

### 3.5. Determination of Minimum Inhibitory Concentrations (MICs) of Enterococcus Strains

The MICs of *Enterococcus* isolates against ampicillin (AMP), ciprofloxacin (CIP), erythromycin (ERY), tetracycline (TET), rifampicin (RIF), chloramphenicol (CHL) and vancomycin (VAN) were determined using the microdilution method recommended by the Clinical and Laboratory Standards Institute (CLSI) [40]. The sensitivity of the strains to antibiotics was categorized as susceptible (S), intermediate (I) or resistant (R) based on CLSI standards.

### 3.6. The Evaluation of the Genetic Variation of Enterococcus Species Through Enterobacterial Repetitive Intergenic Consensus Sequence PCR (ERIC-PCR)

The clonal relatedness of *Enterococcus* species was analyzed via ERIC-PCR reactions in a total volume of 25 µL with the ERIC1-F/R primers according to the protocol described before [41]. The PCR products were validated via electrophoresis on a 3% agarose gel prepared with 5× TBE buffer. Electrophoresis was performed at 90 volts for 4 h, with the gel stained using ethidium bromide. The DNA bands were visualized using an ultraviolet (UV) transilluminator (Alliance 4.7, A XD-79.WL/26MX, Paris, France). The phylogenetic trees were constructed through GelJ software (version 2.0, Logroño, Spain) and visualized by using iTOL https://itol.embl.de/ (accessed on 7 February 2025).

### 3.7. Whole Genome Sequencing, Amplification Analysis and Antibiotic Resistance Gene (ARG) Detection

The erythromycin-resistant *Enterococcus* strains ER01 and erythromycin-susceptible *Enterococcus* ES03 isolated from mice in the HS group and CON group, respectively, were used for whole genome sequencing. Genomic DNA was extracted using a bacterial DNA kit (D3350-01, Omega Bio-tek), and sequencing was conducted using HiSeq technology [42]. The sequence reads were assembled into contigs using SOAPdenovo and annotated using bakta in the Proksee system https://proksee.ca (accessed on 3 February 2025) and BLAST https://blast.ncbi.nlm.nih.gov/Blast.cgi (accessed on 3 February 2025) [43]. Mobile genetic elements (MGEs) were identified by an ICE finder https://bioinfo-mml.sjtu.edu.cn/ICEfinder/ICEfinder.html (accessed on 3 February 2025) and ISfinder https://www-is.biotoul.fr/index.php (accessed on 3 February 2025). Sequence alignment was conducted by using Easyfig 2.2.2 software [44].

An inverse PCR method was used to detect the circular intermediate form of the *ermB*-containing IS*Lgar5* composite transposon and the ICE with the primers listed in Appendix A, and all PCR products were subjected to Sanger sequencing.

The genomic DNA of all isolated *Enterococcus* strains was used as a template for PCR detection of ARGs. Resistance genes, including erythromycin resistance genes (*ermA*, *ermB* and *mefA*), chloramphenicol resistance genes (*fexA* and *optrA*) and tetracycline resistance genes (*tetS* and *tetL*), were detected with the primers listed in Appendix A.

### 3.8. Transfer Experiments

Conjugation transfer was performed with a filter mating assay as demonstrated in previous methods, with minor modifications [45]. The erythromycin-resistant *Enterococcus casseliflavus* strain FZMF (erythromycin resistant but rifampicin susceptible) reported earlier and *Enterococcus avium* ER01 (erythromycin resistant but rifampicin susceptible) in this study served as donors, and the *Enterococcus faecalis* strain ES03 (rifampicin resistant but erythromycin susceptible) in this study served as the recipient [9]. Selection of the transconjugants was performed on Brain Heart Infusion Agar (BHI) (Beijing Solarbio Science & Technology Co., Ltd., Beijing, China) plates containing 256 µg/mL rifampicin and 256 µg/mL erythromycin. Colonies that grew on the plates were further confirmed by the presence of *ermB* and *isa(A)* genes via PCR (Appendix A). The transfer frequency was calculated as the number of transconjugants per donor. Each assay was performed three times.

### 3.9. Intestinal Colonization with Erythromycin-Resistant Enterococcus and Heat Stress in Mice

The colonization assay was performed according to previous reports with slight modifications [46]. Briefly, 60 mice were provided with standard irradiated feed and autoclaved ultrapure water, with free access to food and water. After a 7-day acclimation period, their drinking water was replaced with streptomycin-containing water (1 g/L) for 24 h to eliminate the natural gut microbiota. Twenty-four hours before gavage, the streptomycin water was replaced with sterile water. Fecal samples of 60 mice were collected and plated on medium containing 64 μg/mL erythromycin to confirm the absence of erythromycin-resistant strains prior to gavage. The *Enterococcus* strain ER01 was resuspended at a concentration of 1 × 10^9^ colony-forming units (CFU) in 1 mL of phosphate buffer saline (PBS), and 100 μL was orally administered to each mouse via gavage. After three consecutive days of gavage, fecal samples were collected and plated on selective medium containing 256 μg/mL erythromycin. The presence of colonies confirmed the successful colonization of strain ER01.

Sixty mice with confirmed ER01 colonization were randomly divided into two groups: a heat stress (HS) group and a control (CON) group. The CON group was maintained at a temperature of 25 °C, with free access to food and water. The HS group was subjected to heat stress at 42 °C for 30 min twice daily at 9:00 a.m. and 3:00 p.m. and then housed under the same conditions as the control group after heat stress. At 0, 3 and 6 h after heat stress exposure, five mice were randomly selected from each group and euthanized aseptically. Tissues from the ileum, cecum, colon and rectum were collected and washed three times with PBS. The intestinal contents were weighed, and PBS was added at a ratio of 1 mg:10 μL. After appropriate serial dilutions, a 20-μL aliquot of the suspension was plated on selective medium with 256 μg/mL erythromycin. The plates were incubated at 37 °C for 48 h. The percentage P was calculated as P = CFU_x_/CFU_t._, where CFU_t_ represents the total CFUs across the entire intestine and CFU_x_ represents the CFUs in a specific intestinal segment (ileum, cecum, colon or rectum).

### 3.10. Metagenomic Sequencing and Analysis

Sterile samples of ileum and rectum contents were collected from the CON group and the HS group after 55 days of exposure, with five mice per group. The groups were labeled as ileum control (HA), ileum heat stress (HB), rectum control (ZA) and rectum heat stress (ZB). DNA extraction, purity assessment and concentration measurements were performed while following the DNA extraction kit’s protocol. DNA was fragmented using an ultrasonic disruptor, and libraries were constructed using a NEXTFLEX Rapid DNA-Seq Kit (Bioo Scientific, Austin, TX, USA). Libraries were sequenced on the Illumina NovaSeq platform (Wuhan IGENEBOOK Biotechnology Co., Wuhan, China) for comprehensive genomic information.

Raw reads were processed with fastp to remove low-quality reads, retaining high-quality reads for reliable analysis [47,48]. Reads were aligned to host DNA sequences using BWA http://bio-bwa.sourceforge.net (accessed on 3 September 2023) to remove contaminating host reads. Assembled data were processed with MEGAHIT, ensuring accuracy and reliability [49,50]. Diamond software https://github.com/bbuchfink/diamond (accessed on 3 September 2023) was used to align amino acid sequences with the NCBI NR database (Version: 2021.11) for species annotation, and species abundances were calculated based on gene abundance [51,52]. KEGG database alignment provided insights into functional gene abundance, while ARDB analysis identified antibiotic resistance functions and abundance, enabling a detailed understanding of the microbial communities’ predicted antibiotic resistance characteristics [53,54,55]. Analysis of similarity (Anosim) was used to test the significance of differences between and within groups based on the species-level microbial composition [56]. Linear discriminant analysis effect size (LefSe) analysis was performed to identify the significant differences in genomic pathway abundance across the groups [57]. To identify ARGs with significant differences between groups, the Metastats method was performed to conduct testing on the abundance data of resistance genes. The *p* values were calculated, and false discovery rate correction was applied to obtain *q* values. Differentially abundant resistance genes were then selected based on the *q* values, and a bar chart of the differentially abundant resistance genes was plotted [58].

## 4. Conclusions

This study found that heat stress significantly affects the microbial composition and its metabolism in mouse intestine, transitioning it toward more anaerobic metabolic processes. The changes in the gut microenvironment promoted the transfer and integration of *ermB*-containing ICE_FZMF_, leading to the emergence of erythromycin-resistant *Enterococcus*. Additionally, the changes in the rectal environment induced by heat stress make it easier for the colonization and proliferation of these erythromycin-resistant strains. These findings explore the mechanisms behind the development of bacterial resistance in the mouse gut under heat stress, providing a reference for healthy animal breeding processes and reducing bacterial resistance.

## Figures and Tables

**Figure 1 antibiotics-14-00460-f001:**
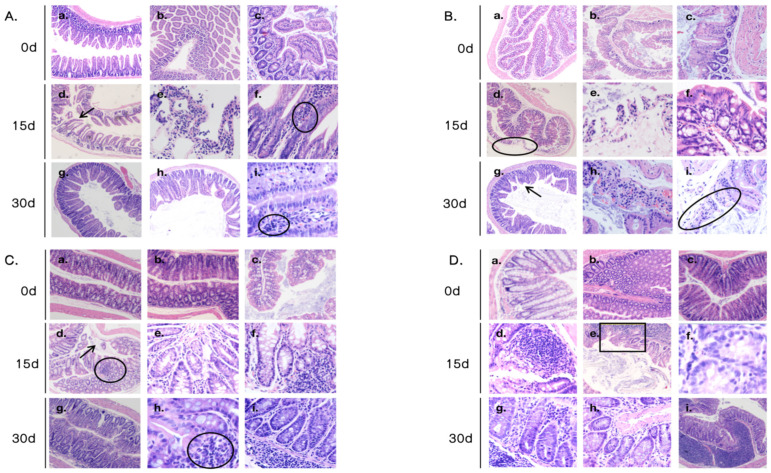
Intestinal morphology. (**A**) Pathological observation of ileum in mice. (**a**–**c**) The ileum tissue of the control group. (**d**–**f**) The ileum tissue after 15 days of heat stress. (**g**–**i**) The ileum tissue after 30 days of heat stress. The arrows indicate necrosis and sloughing of the ileal mucosa in (**d**), and the inflammatory cell infiltration is circled in (**f**,**i**). (**B**) Pathological observation of cecum in mice. (**a**–**c**) The cecal tissue of the control group. (**d**–**f**) The cecal tissue after 15 days of heat stress. (**g**–**i**) The cecal tissue after 30 days of heat stress. The apoptotic necrosis of the cecal mucosal glands and glandular atrophy are circled in (**d**). The arrow points to the glandular atrophy in (**g**), and the apoptotic necrosis and stromal edema are circled in (**i**). (**C**) Pathological observation of colon in mice. (**a**–**c**) The colonic tissue of the control group. (**d**–**f**) The colonic tissue after 15 days of heat stress. (**g**–**i**) The colonic tissue after 30 days of heat stress. The arrow points to the glandular atrophy, and the lymphoid follicle proliferation is circled in (**d**), while the lymphoid follicle is circled in (**h**). (**D**) Pathological observation of rectum in mice. (**a**–**c**) The rectal tissue of the control group. (**d**–**f**) The rectal tissue after 15 days of heat stress. (**g**–**i**) The rectal tissue after 30 days of heat stress. The glandular atrophy is framed in (**e**).

**Figure 2 antibiotics-14-00460-f002:**
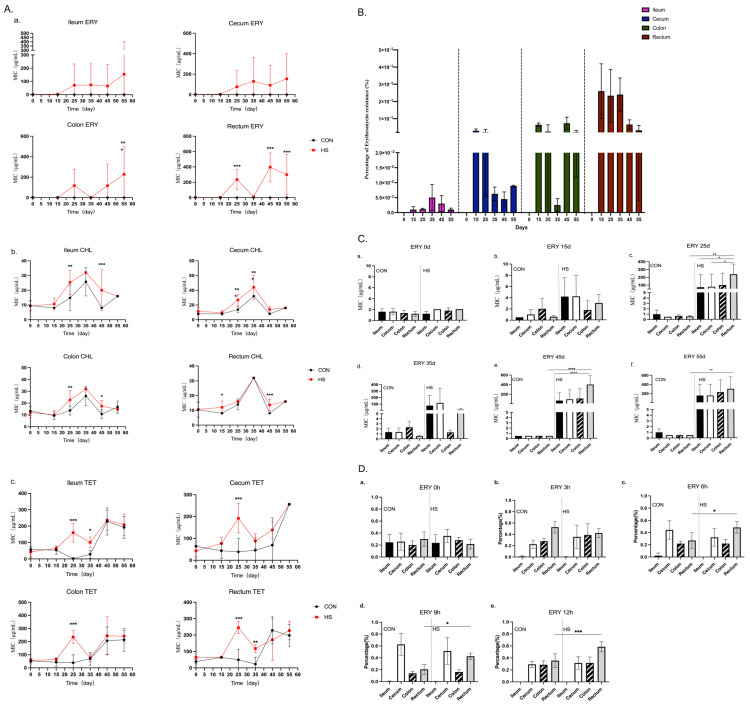
The increased erythromycin resistance among *Enterococcus* strains and the colonization of erythromycin-resistant strains. (**A**) The MIC values of erythromycin, chloramphenicol and tetracycline among strains: (**a**) The changes in MIC values of erythromycin among strains collected from different intestinal segments, (**b**) The changes in MIC values of chloramphenicol among strains collected from different intestinal segments, (**c**) the changes in MIC values of tetracycline among strains collected from different intestinal segments. Asterisks show significant differences from the control group using two-way ANOVA with Bonferroni’s multiple comparison test (*** *p* < 0.001, ** *p* < 0.01, * *p* < 0.05 and not significant (ns)). (**B**) The rate of erythromycin-resistant strains in diverse intestinal segments. (**C**) The erythromycin MIC values of strains isolated from diverse segments at day 0 (**a**), day 15 (**b**), day 25 (**c**), day 35 (**d**), day 45 (**e**) and day 55 (**f**). These were compared using one-way ANOVA (**** *p* < 0.0001, ** *p* < 0.01 and * *p* < 0.05). (**D**) The colonization of erythromycin-resistant strain ER01 at 0 h (**a**), 3 h (**b**), 6 h (**c**), 9 h (**d**) and 12 h (**e**). Data were compared using one-way ANOVA (*** *p* < 0.001 and * *p* < 0.05).

**Figure 3 antibiotics-14-00460-f003:**
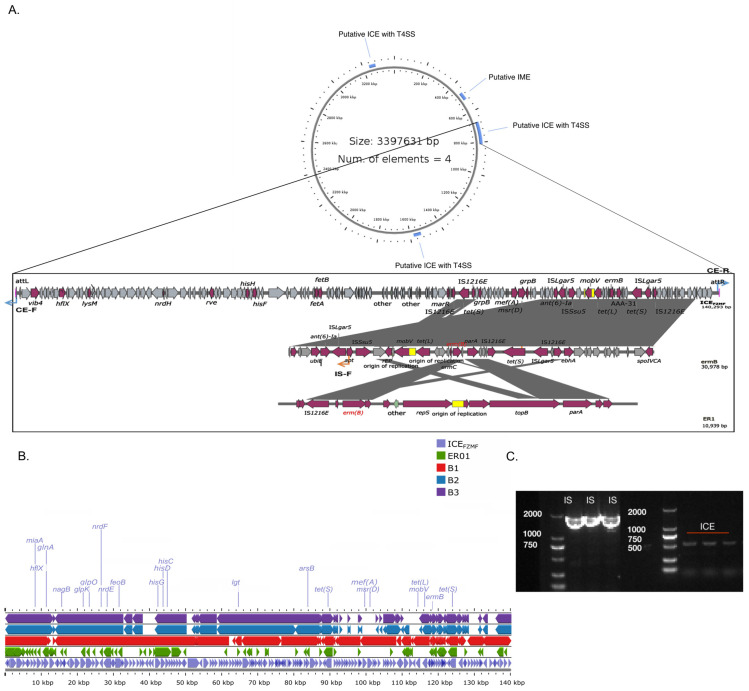
The sequence of ICE_FZMF_. (**A**) The four putative ICEs are displayed in the upper part, and the organization of ICE_FZMF_ is shown on the first line of the bottom part. The surrounding region of *ermB* in ICE_FZMF_ is shown on the second line, and the *ermB* surrounding region in the ER01 is present on the third line. The arrow indicates the CE-F/R and IS-F/R primers for confirming the formation of a circular intermediate. IME is the abbreviation for integrative and mobilizable element. (**B**) The highly similar sequences of ER01, B1, B2 and B3 are mapped to the ICE_FZMF_ sequence. (**C**) PCR results for confirming the formation of a circular intermediate.

**Figure 4 antibiotics-14-00460-f004:**
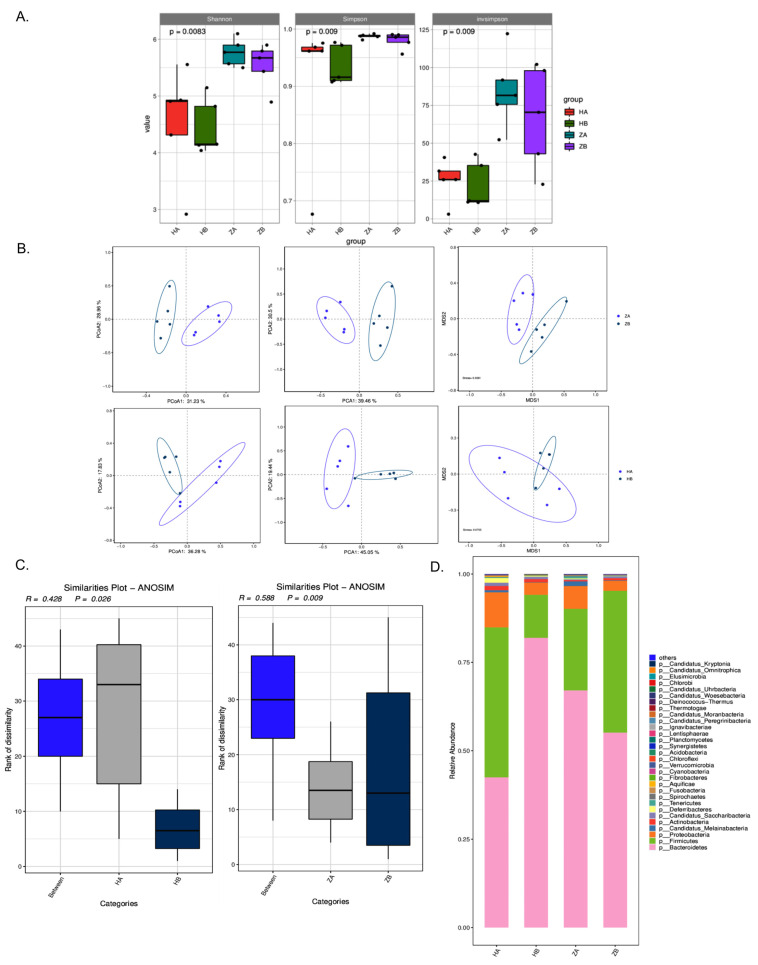
Analysis of the microbial composition in the ileum and rectum. (**A**) The α diversity analysis of the microbial composition. (**B**) The β diversity analysis of the microbial composition. (**C**) Anosim analysis based on species level. (**D**) Relative species abundance of gut microbes at phylum level. HA indicates the ileum sample without heat stress, HB indicates the ileum sample after heat stress. ZA indicates the rectal sample without heat stress, ZB indicates the rectal sample after heat stress.

**Figure 5 antibiotics-14-00460-f005:**
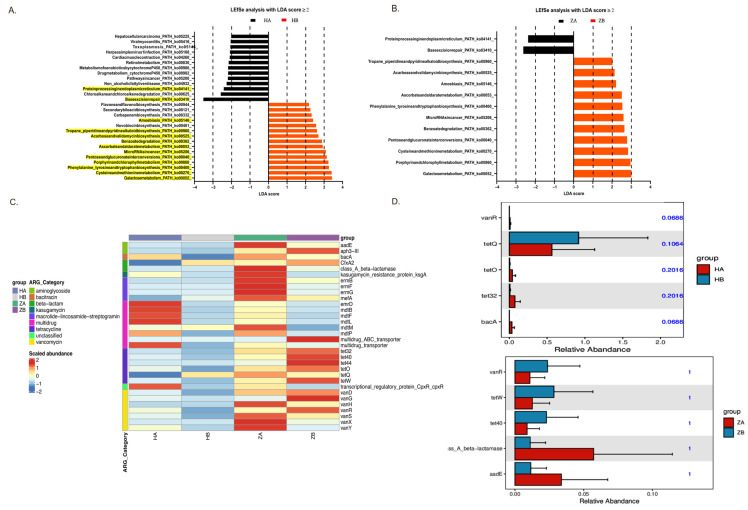
The Metascape and antibiotic-resistance gene enrichment analysis. (**A**,**B**) The effect of heat stress on KEGG level 3 function in intestinal microbes under LEfSe analysis. LDA scores ≥2 are displayed. The pathways that showed similar trends in both the ileum and rectum are highlighted in yellow in (**A**). (**C**) Heat map of intestinal resistance gene abundance induced by heat stress in mice. (**D**) Abundance of resistance genes in the ileum and rectum of mice induced by heat stress. HA indicates the ileum sample without heat stress, and HB indicates the ileum sample after heat stress. ZA indicates the rectal sample without heat stress, and ZB indicates the rectal sample after heat stress.

**Table 1 antibiotics-14-00460-t001:** Detection rates of antimicrobial resistance genes among isolates (%).

Groups	Segments	*ermA*	*ermB*	*mefA*	*fexA*	*optrA*	*tetS*	*tetL*
CON	Ileum	0.0 (0/32)	0.0 (0/32)	0.0 (0/32)	0.0 (0/32)	0.0 (0/32)	0.6 (20/32)	0.1 (2/32)
	Cecum	0.0 (0/48)	0.0 (0/48)	0.0 (0/48)	0.0 (0/48)	0.0 (0/48)	0.3 (13/48)	0.1 (3/48)
	Colon	0.0 (0/34)	0.0 (0/34)	0.0 (0/34)	0.0 (0/34)	0.0 (0/34)	0.5 (18/34)	0.1 (4/34)
	Rectum	0.0 (0/40)	0.0 (0/40)	0.0 (0/40)	0.0 (0/40)	0.0 (0/40)	0.5 (20/40)	0.02 (1/40)
	Total	0.0 (0/154)	0.0 (0/154)	0.0 (0/154)	0.0 (0/154)	0.0 (0/154)	0.5 (71/154)	0.1 (10/154)
HS	Ileum	2.1 (1/48)	31.2 (15/48)	4.2 (0/48)	2.1 (1/48)	0.0 (0/48)	0.3 (12/48)	0.02 (1/48)
	Cecum	0.0 (0/55)	58.2 (32/55)	1.8 (0/55)	10.9 (6/55)	1.8 (1/55)	0.4 (20/55)	0.1 (5/55)
	Colon	0.0 (0/62)	68.8 (44/62)	1.6 (0/62)	1.6 (1/62)	1.6 (1/62)	0.5 (30/62)	0.1 (4/62)
	Rectum	1.7 (1/60)	76.7 (46/60)	3.3 (1/60)	5 (3/60)	1.7 (1/60)	0.3 (15/60)	0.02 (1/60)
	Total	0.9 (2/225)	60.1 (137/225)	0.5 (1/225)	4.9 (11/225)	1.3 (3/225)	0.3 (77/225)	0.05 (11/225)

## Data Availability

The complete nucleotide sequences of erythromycin-resistant *Enterococcus* isolate ER01 and erythromycin-susceptible *Enterococcus* isolate ES03 have been deposited at GenBank (PRJNA1231587). The Sanger sequence for confirming the formation of a circular intermediate has been deposited at Genbank (PRJNA1231627).

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
