# Peer review of "Chronic Heat Stress Can Induce Conjugation of a Novel ermB-Containing ICEFZMF, Increasing Resistance to Erythromycin Among Enterococcus Strains in Diverse Intestinal Segments in the Mouse Model"

_antibiotics, 2025, doi:10.3390/antibiotics14050460_

Round 1

Reviewer 1 Report

Comments and Suggestions for Authors

The manuscript entilted "Chronic heat stress can induce conjugation of a novel ermB-containing ICEFZMF, increasing resistance to erythromycin among Enterococcus strains in diverse intestinal segments in mouse model" by Yi and co-authors describe experiments where mice were exposed to heat stress conditions and the effects of that stress on the erythromycin resistance  of Enterococcus strains in the various segments of the intestine. The authors subjected mice to heat stress at 42ºC twice a day for 30 minutes hour and analyzed several parameters such as the  production of  mRNA encoding heat shock proteins, markers of inflamation, and intestine integrity by microscopic  analysis of tissues. The authors also sampled and isolated bacteria from various sections of the intestinal track of mice exposed to heat stress and controls, and analysed the MIC values for Enterococcus strains, concluding that heat stress predisposed animals to colonization by erytromycin resistant enterococci. The manuscript is fairly organized, and no major issues were detected concerning the english language usage. There are however some issues tha tshould be addressed by the authors:

1) The Abstract should be re-written to include the description of the establishment of a heat stress mice model.

2) Results and Discussion section: The authors state that a heat stress model ws established. In this view, the sections 3.1 and 3.2 should be titled according to this, and not as they are titled now. 

3) Figures 2, 3, 4 and 5 are impossible to read and should be enlarged in size, othe wise no one can read them. 

4) Table 1 needs to be fixed in the last lines, withou  brackets appearing alone, and should appear in one page, not broken into 2 pages.

5) Fig. S8 is missing from the supplementary material, I could not see it on the suplementary material.

Other issues:

line 13: define SPF

line 72: heat stimulation?

line 84: Define ICR

line 87: write as fllows:"...., food and water..."

line 102: which extraction kit?

line 107: How histological sections were prepared  and which procedures were used to visualize and take pictures shown in Fig. 2? what is the magnification in Fig 2?

lines 386-391 - Is the ERIC-PCR technique good enough to establish phylogenetic relationships?

line 345: ...control group were all sensitive...

line 437: confirmation of the circular intermediation? Where is this confirmation?

lines 471 -472: define ZA and ZB

Author Response

  • The Abstract should be re-written to include the description of the establishment of a heat stress mice model.

A: Thanks for your suggestion. We have revised the abstract to briefly describe the heat stress model, including the temperature, frequency and duration of exposure in Line 12-32. We also confirm that the revised abstract contains fewer than 250 words, which is accordance with the journal’s requirements.

  • Results and Discussion section: The authors state that a heat stress model ws established. In this view, the sections 3.1 and 3.2 should be titled according to this, and not as they are titled now. 

A: Thanks for your suggestion, the section titles for 3.1 and 3.2 have been revised in Line 225 and 295, focusing on model establishment and validation. Also, we add small edits in line 253-254, 301, that could help clarify the role of these sections in model validation.

  • Figures 2, 3, 4 and 5 are impossible to read and should be enlarged in size, othe wise no one can read them. 

A: Thanks for your comments, Figures 2-5 have now been reformatted and enlarged to better visualization.

  • Table 1 needs to be fixed in the last lines, withou  brackets appearing alone, and should appear in one page, not broken into 2 pages.

A: Thanks for your comments, we have revised accordingly in Line 392.

  • S8 is missing from the supplementary material, I could not see it on the suplementary material.

A: Thanks for your careful observation. We apologize for the confusion. Figure S8 was combine into Figure S7 as panel B in the supplementary material. So, the reference in Line 479 should be Figure S7B. We have revised the manuscript accordingly. 

  • line 13: define SPF

A: Thanks for your comments, “SPF” has been defined as “specific pathogen Free” upon first mention in line 14.

  • line 72: heat stimulation?

A: Thanks for your comments, we have revised the term ‘heat stimulation” to “heat stress” for consistency and accuracy in Line 74

  • line 84: Define ICR

A: Thanks for your comments, the full name of ICR mice is SLAC:ICR, a strain of outbred Swiss albino mice commonly used in safety, efficacy evaluations and surgical models. we have revised accordingly in Line 85.

  • line 87: write as fllows:"...., food and water..."

A: Thanks for your comments, we have revised accordingly in line 88.

  • line 102: which extraction kit?

A: Thanks for your comments, we have revised accordingly in line 113.

  • line 107: How histological sections were prepared  and which procedures were used to visualize and take pictures shown in Fig. 2? what is the magnification in Fig 2?

A:Thanks for the comments, we have added the description of the histological section preparation and staining procedures, including tissue dehydration, paraffin embedding, sectioning, H&E staining, and slide Imagin in Line 95-108.

  • lines 386-391 - Is the ERIC-PCR technique good enough to establish phylogenetic relationships?

A: Thanks for your comments, we agree that Eric-PCR is not appropriated for precise phylogenetic analysis. our aim in this section was to assess the clonal relatedness of the erythromycin-resistant Enterococcus strains, rather than to perform a comprehensive phylogenetic study. So, we revise the “phylogenetic analysis” into “clonal typing” in Line 411 ,and the conclusion focuses on the genetic diversity and non-clonal of these strains, we also revised the section title in Line 380.

  • line 345: ...control group were all sensitive...

A: Thanks for your comments, actually, the enterococci isolate from the control group already exhibited resistance to chloramphenicol and tetracycline. They are not all sensitive. For more clarification, I add the sentence” although the tetracycline- and chloramphenicol- resistant isolates were also detected in the control group, their overall MIC distributions were lower than those observed in the heat stress group.” in Line 370-372.

  • line 437: confirmation of the circular intermediation? Where is this confirmation?

A: Thanks for your comments, the formation of a circular intermediate of ICEFZMF and ISLgar5-tet(L)-ermC-ermB-tet(S)-ISLgar5 was experimentally validated by inverse PCR, followed by Sanger sequencing with primer CE-F/R and IS-F/R. the primers were listed in Supplementary materials. The location of these primers was also indicated in Figure 3A. The PCR results were showed in Figure 3C. To improve clarity, we have now specified this confirmation more explicitly in both the results and methods sections in Line 428-430,435-438, 443. The sanger sequence has also been deposited in the NCBI database under assession number: PRJNA1231627.

  •  lines 471 -472: define ZA and ZB

A: Thank you for your comments, ZA and ZB were defined in line 209-210.

Reviewer 2 Report

Comments and Suggestions for Authors

I have reviewed the manuscript Chronic Heat Stress Can Induce Conjugation of a Novel
ermB-Containing ICEFZMF, Increasing Resistance to Erythromycin Among Enterococcus
Strains in Diverse Intestinal Segments in Mouse Model.
In this study, the impact of heat stress on antibiotic susceptibility and intestinal functions
were tested in the mouse model. To this aim, mice were subjected to heat stress (42°C)
twice daily, and the control mice were kept at room temperature at all times. Mice
intestines were extensively researched. In addition to damage to the intestines, heat stress
was found to be associated with increased resistance to chloramphenicol, tetracycline, and
erythromycin.
Major concern
Ethical clearance is not mentioned in the manuscript. Has ethical clearance been
obtained?
Other comments and suggestions
General comments
Abstract:
Please describe more clearly how many mice were included in each experiment. Also, not
all HS mice were subjected to 42°C for 1 hour, daily for 55 days. Please add more details,
or alterntively just describe what was the case for all the mice.
Discussion:
It may be of interest to discuss the measures that can be taken to avoid heat stress in
livestock animals.
In Switzerland, for example, livestock is often at different altitudes depending on the
season of the year. Below is an example of a manuscript that suggests other ways to
address heat stress: Adaptation to hot climate and strategies to alleviate heat stress in
livestock production 10.1017/S1751731111002448.
Specific comments:
Lines 4-5: Change to “… Segments in the Mouse Model”, adding the article “the”.
Line 51: Change to “Enterococci are widely prevalent opportunistic pathogens, playing a
crucial role …”
Lines 14-15 and 88-89: Were the HS group mice exposed to heat stress for one hour twice
daily or for 30 minutes twice daily? Lines 15, 91, and 110 seem contradictory. Please
clarify when the mice were killed for the harvesting of intestines (you might want to add
how many mice were killed each time). Please describe the difference between mouse
experiments more clearly.
Line 93: Change to “dried”.
Line 161: Change “weas” to “was”.
Line 180: The abbreviation for control is “CN” altough it was “CON” previously. Is this
intentional?
Line 292: Change to “impair”.
Line 296: Change to “confirmation”.
Lines 461-462: Rephrase “The intestine is frequently regarded as the reservoir for the
cultivation of bacterial and its development of antibiotic resistance”.

Author Response

  • Ethical clearance is not mentioned in the manuscript. Has ethical clearance been obtained?

A: Thanks for pointing this out, we confirm the Ethics Committee approval was obtained for the animal experiments. This work has received ethical approval from the Research Ethics Committee of institute of Animal Science of Fujian Agriculture and Forestry University (No. PZCASFAFU24009), we also add the related content in Line 595-597

  • Please describe more clearly how many mice were included in each experiment. Also, not all HS mice were subjected to 42°C for 1 hour, daily for 55 days. Please add more details,or alterntively just describe what was the case for all the mice.

A: Thank you for your Comment. We have now revised to clearly describe the experimental design, including the number of mice used, treatment conditions, and the time points of sampling in Line 85-114,121-123. All mice in the HS group were subjected to a consistent heat stress regimen of 42°C for 1 hour daily (30 minutes in the morning and 30 minutes in the afternoon) for 55 consecutive days, unless euthanized at intermediate time points for sampling. I think the original abstract caused the confusion; we have revised it in Line 15.

  • Discussion:
    It may be of interest to discuss the measures that can be taken to avoid heat stress in
    livestock animals.
    In Switzerland, for example, livestock is often at different altitudes depending on the
    season of the year. Below is an example of a manuscript that suggests other ways to
    address heat stress: Adaptation to hot climate and strategies to alleviate heat stress in
    livestock production 10.1017/S1751731111002448.

A: Thanks for your comments, we have included a discussion in the manuscript about the potential measure to prevent the heat stress in livestock animals in Line 570-579

  • Lines 4-5: Change to “… Segments in the Mouse Model”, adding the article “the”.

A: Thank you for your comments, we have changed the title accordingly in Line 5.

  • Line 51: Change to “Enterococci are widely prevalent opportunistic pathogens, playing a
    crucial role …”

A: Thanks for your comments, we have revised accordingly in line 52.

  • Lines 14-15 and 88-89: Were the HS group mice exposed to heat stress for one hour twice
    daily or for 30 minutes twice daily?

A: Thanks for your comments, it is 30 minutes twice daily, totally 1 h daily, I have revised it in Line15.

  •  Lines 15, 91, and 110 seem contradictory. Please clarify when the mice were killed for the harvesting of intestines (you might want to add how many mice were killed each time). Please describe the difference between mouse experiments more clearly.

A: Thank you for your insightful comment. We have clarified the experimental design by specifying the exact number of mice used for each type of analysis (bacterial isolation, histology, RT-qPCR, and metagenomic sequencing), as well as the time points at which mice were sacrificed. This information has been incorporated in line 95-96,111 and 121-122.

Briefly, Bacterial isolation: Days 0, 15, 25, 35, 45, and 55 – 3 mice per group per time point, intestinal contents collected. we hope to know the MIC change in detail, so we set more time point for the assays.

Histological analysis (HE staining): Days 0, 15, and 30 – 2 mice per group per time point, intestinal tissue fixed in paraformaldehyde.

qPCR analysis: Days 7, 15, 30, and 45 – 5 mice per group per time point, tissue used for RNA extraction.

Metagenomic sequencing: Day 55 – 5 mice per group, intestinal contents collected.

  • Line 93: Change to “dried”.

A: Thanks for your comments, since we rewritten the method, the word has been deleted.

  • Line 161: Change “weas” to “was”.

A: Thanks for your comments, we have revised accordingly in line 174.

  • Line 180: The abbreviation for control is “CN” altough it was “CON” previously. Is this
    intentional?

A: We apologize for the inconsistency. we have revised the manuscript to ensure the abbreviation for the control group is consistently written as “CON” throughout the manuscript. (Line 193)

  • Line 292: Change to “impair”.

A: Thanks for your comments, we have revised accordingly in line 306.

  • Line 296: Change to “confirmation”.

A: Thanks for your comments, we have revised accordingly in line 311.

  • Lines 461-462: Rephrase “The intestine is frequently regarded as the reservoir for the
    cultivation of bacterial and its development of antibiotic resistance”.

A: Thanks for your comments, we have revised accordingly in line 485-486.

Round 2

Reviewer 1 Report

Comments and Suggestions for Authors

The revised version of the manuscript correctly addresses the criticisms raised to the previous version. No further criticisms are raised by this reviewer.

Author Response

Thanks